



# KINETICS OF CALCITE PRECIPITATION BY UREOLYTIC BACTERIA UNDER AEROBIC AND ANAEROBIC CONDITIONS

Andrew C. Mitchell[1,2], Erika J. Espinosa-Ortiz[2], Stacy L. Parks[2,3], Adrienne Phillips[2,4], Alfred B.
Cunningham[2,4], Robin Gerlach[2,3]
[1]Department of Geography and Earth Sciences, Interdisciplinary Centre for Environmental Microbiology,
Aberystwyth University, SY23 3DB, UK.
[2]Center for Biofilm Engineering, Montana State University, Bozeman, MT, 59717, USA.
[3]Department of Chemical and Biological Engineering, Montana State University, Bozeman, MT 59717, USA.
[4]Department of Civil Engineering, Montana State University, Bozeman, MT 59717, USA.
*Correspondence to*: Andrew C. Mitchell (nem@aber.ac.uk), and Robin Gerlach (robin_g@coe.montana.edu)
**Key words:** $CaCO_3$, carbonate precipitation, ureolysis, biomineralization, kinetics, geologic carbon capture and
storage.





**Abstract**. The kinetics of urea hydrolysis (ureolysis) and induced calcium carbonate ($CaCO_3$) precipitation for
engineering use in the subsurface was investigated under aerobic conditions using *Sporosarcina pasteurii* (ATCC
strain 11859) as well as *Bacillus sphaericus* strains 21776 and 21787. All bacterial strains showed ureolytic activity
inducing $CaCO_3$ precipitation aerobically. Rate constants not normalized to biomass demonstrated slightly higher rate
coefficients for both ureolysis ($k_{urea}$) and $CaCO_3$ precipitation ($k_{precip}$) for *B. sphaericus* 21776 ($k_{urea} = 0.10 \pm 0.03$ $h^{-1}$,
$k_{precip} = 0.60 \pm 0.34$ $h^{-1}$) compared to *S. pasteurii* ($k_{urea} = 0.07 \pm 0.02$ $h^{-1}$, $k_{precip} = 0.25 \pm 0.02$ $h^{-1}$). *B. sphaericus* 21787
showed little ureolytic activity but was still capable of inducing some $CaCO_3$ precipitation. Cell growth appeared to
be inhibited during the period of $CaCO_3$ precipitation. TEM images suggest this is due to the encasement of cells and
was reflected in lower $k_{urea}$ values observed in the presence of dissolved Ca. However, biomass re-growth could be
observed after $CaCO_3$ precipitation ceased, which suggests that ureolysis-induced $CaCO_3$ precipitation is not
necessarily lethal for the entire population. The kinetics of ureolysis and $CaCO_3$ precipitation with *S. pasteurii* were
further analyzed under anaerobic conditions. Rate coefficients obtained in anaerobic environments were comparable
to those under aerobic conditions, however no cell growth was observed under anaerobic conditions with $NO_3^-$, $SO_4^{2-}$
and $Fe^{3+}$ as potential terminal electron acceptors. These data suggest that the initial rates of ureolysis and ureolysis-
induced $CaCO_3$ precipitation are not significantly affected by the absence of oxygen but that long-term ureolytic
activity might require the addition of suitable electron acceptors. Such variations in the ureolytic capabilities and
associated rates of $CaCO_3$ precipitation between strains must be fully considered in subsurface engineering strategies
that utilize microbial amendments.



## 1. Introduction

Carbonate precipitation is a natural phenomenon which may also be utilized in many subsurface engineering applications (Phillips et al., 2013a) including soil stabilization (van Paassen Leon et al., 2010), immobilization of radionuclides (Mitchell and Ferris, 2005, 2006;Tobler et al., 2012;Warren et al., 2001), and mineral plugging for enhanced oil recovery and carbon sequestration (Dupraz et al., 2009;Ferris et al., 1996;Mitchell et al., 2010;Phillips et al., 2013b). Mineral precipitation can be induced by bacteria as a by-product of common microbial processes, such as urea hydrolysis (ureolysis). In this process, bacteria hydrolyze urea ($CO(NH_2)_2$), an important nitrogen compound found in natural environments) to ammonia ($NH_3$) and carbonic acid ($H_2CO_3$) (Equations 1-3). The $NH_3$ and $H_2CO_3$ equilibrate in circumneutral aqueous environments to form bicarbonate ($HCO_3^-$), two ammonium ions ($NH_4^+$) and one hydroxide ion ($OH^-$) (Equations 4-5), or at higher pH values to one carbonate ion ($CO_3^{2-}$) and two $NH_4^+$ (Equations 4-6). In the presence of dissolved calcium ($Ca^{2+}$), this increase in carbonate alkalinity shifts the saturation state of the system, allowing for solid calcium carbonate ($CaCO_3$) to form (Equation 7). The overall reaction from the hydrolysis of urea in the presence of $Ca^{2+}$ is summarized by Equation 8.

$$CO(NH_2)_2 + H_2O \rightarrow NH_2COOH + NH_3 \tag{1}$$

$$NH_2COOH + H_2O \rightarrow NH_3 + H_2CO_3 \tag{2}$$

$$CO(NH_2)_2 + 2H_2O \rightarrow 2NH_3 + H_2CO_3 \text{ (Equations 1 + 2 overall)} \tag{3}$$

$$2NH_3 + 2H_2O \leftrightarrow 2NH_4^+ + 2OH^- \tag{4}$$

$$H_2CO_3 \leftrightarrow HCO_3^- + H^+ \tag{5}$$

$$HCO_3^- + H^+ + 2OH^- \leftrightarrow CO_3^{2-} + 2H_2O \tag{6}$$

$$CO_3^{2-} + Ca^{2+} \leftrightarrow CaCO_3 \tag{7}$$

$$CO(NH_2)_2 + 2H_2O + Ca^{2+} \leftrightarrow 2NH_4^+ + CaCO_3 \text{ (Overall process)} \tag{8}$$

The use of ureolytic bacteria in biotechnological applications is appealing for many reasons. Ureolytically active microorganisms are common in a wide variety of soil and aquatic environments, thus, indigenous microorganisms capable of ureolysis can be either stimulated *in situ* or alternatively, they can be used to augment environments lacking ureolytic microorganisms (Fujita et al., 2000;Warren et al., 2001). Urea is a fairly inexpensive substrate and it is often contained in wastewater (Hammes et al., 2003b), so this waste product may be used to stimulate ureolysis in engineering applications (Mitchell et al., 2010). Moreover, the controlled increase of pH and alkalinity in the subsurface by ureolytic bacteria is preferable to the injection of a basic solution (abiotic process), which could lead to instantaneous $CaCO_3$ supersaturation and precipitation at the point of injection limiting the radius of influence of the technology. The injection of urea into the subsurface followed by microbially induced ureolysis would allow for the controlled, gradual ureolysis further away from the injection point, promoting a wider spatial distribution of $CaCO_3$ in the subsurface and avoiding uncontrolled plugging at the point of injection (Cuthbert et al., 2013;Ebigbo et al., 2012;Mitchell and Ferris, 2005;Schultz et al., 2011;Tobler et al., 2012;Tobler et al., 2014).

Among different ureolytic bacteria, *Sporosarcina pasteurii* (formerly known as *Bacillus pasteurii*) has been extensively used as the model urease-producing organism in ureolysis-driven $CaCO_3$ precipitation studies due to its




high ureolytic activity and constitutive production of urease (Phillips et al., 2013). The use of *S. pasteurii* for $CaCO_3$
precipitation is feasible under aerobic conditions and the kinetics of ureolysis under different conditions have been
studied. Most studies have reported first order ureolysis rates with respect to urea concentration, with the rate constant
($k_{urea}$) ranging between 0.002 and 0.090 $h^{-1}$ under aerobic conditions in artificial groundwater without nutrients added
(Dupraz et al., 2009;Ferris et al., 2004;Hammes et al., 2003a;Mitchell and Ferris, 2005;Tobler et al., 2012), and 0.35
$h^{-1}$ with the addition of nutrients (Lauchnor et al., 2015). Ureolysis rates have been suggested to be temperature-
dependent (Ferris et al., 2004), and it seems to also be affected by cell concentration (inoculum size) (Lauchnor et al.,
2015;Tobler et al., 2011).
Although the ureolytic activity of *S. pasteurii* under anoxic conditions has been observed (Martin et al.,
2012;Mortensen et al., 2011;Tobler et al., 2012), there is controversy regarding the extent and duration of ureolytic
activity that can be achieved in the absence of oxygen. Mortensen et al. (2011) and Tobler et al. (2012) observed
extensive ureolytic activity under anoxic conditions, suggesting that the anoxic environment does not inhibit urease
activity. Conversely, Martin et al. (2012) observed limited cell growth and poor ureolysis under anoxic conditions and
suggested that the ureolytic activity observed was due to the urease already present in the cells.
In this study, the ability of *S. pasteurii* to grow in the absence of oxygen (with or without nitrate ($NO_3^-$), sulfate
($SO_4^{2-}$) or ferric ion ($Fe^{3+}$) as possible electron acceptors) was investigated along with the kinetics of ureolysis and
$CaCO_3$ precipitation. Moreover, this study investigates and compares the ureolytic activity of *S. pasteurii* with
different strains of *Bacillus sphaericus* under aerobic conditions, which have also been suggested to be capable of
ureolysis-induced $CaCO_3$ precipitation (Dick et al., 2006;Hammes et al., 2003a).
**2. Materials and methods**
**2.1. Solutions**
Kinetic experiments were carried out using the $CaCO_3$ Mineralizing Medium (CMM) described by Ferris and
Stehmeier (1996) (see Supplemental Information [SI], Table SI1.1). Both $Ca^{2+}$ inclusive (CMM+) and $Ca^{2+}$ exclusive
(CMM-) versions of this medium were used. Aerobic CMM- was prepared as follows. A double strength solution of
nutrient broth was prepared and autoclaved. A nutrient broth was chosen in this experiment to enable cell growth. A
separate solution of double strength urea, ammonium chloride, and sodium bicarbonate was prepared and stirred until
completely dissolved. These two solutions were combined and adjusted to a pH of 6.0 using concentrated HCl. CMM+
was prepared similarly, but calcium chloride was added after the pH adjustment. Media were filter sterilized into
sterile Pyrex bottles using 0.2 μm pore size filters (Nalgene, Rochester, NY). Anaerobic CMM was produced in the
same manner. However, all stock solutions were made in an anaerobic chamber using water that had been degassed
by stirring overnight in the oxygen-free atmosphere of the chamber (90% $N_2$, 5% $CO_2$, 5% $H_2$). Solutions were filter
sterilized into serum bottles and were then capped and sealed inside the chamber. Prior to experiments, the solutions
were combined to reach the final concentrations listed in Table SI1.1.



### 2.2. Bacterial strains and culturing conditions

Three strains of ureolytic bacteria were used: *S. pasteurii* (ATTC 11859), and two isolates from a garden soil and landfill soil, *B. sphaericus* 21776, and *B. sphaericus* 21787 (Belgian Coordinated Collections of Microorganisms, Laboratory of Microbiology, Ghent University) (Hammes et al., 2003b). *Bacillus subtilis* strain 186 (ATCC 23857), a non-ureolytic organism, was used as a control species. Abiotic (i.e. non-inoculated) controls were also set up and run in parallel. Pilot cultures were grown in flasks by adding 100 μL of thawed stock to 100 mL of autoclaved Brain Heart Infusion (BHI) + 2% urea. *S. pasteurii* and *B. sphaericus* were grown on an incubator shaker (New Brunswick Scientific, Edison, NJ) at 30°C and 150 rpm, while *B. subtilis* was grown on an incubator shaker at 37°C and 150 rpm. 100 μL of pilot cultures were transferred at 24 h and 48 h to new flasks containing 100 mL of BHI + 2% urea.

### 2.3. Aerobic experiments

Once the pilot cultures were ready for inoculation, 40 mL of culture were added to a 50 mL centrifuge tube. This tube was centrifuged at $4303 \times g$ using a Sorvall Instruments (Asheville, NC) RC-5C centrifuge for 10 min at 4-6°C. The supernatant was poured off the cell pellet, and it was re-suspended using about 40 mL of CMM-, and again centrifuged for 10 min. This process was repeated once more. After the third run in the centrifuge, the supernatant was poured off and enough CMM- was added to achieve a final optical density reading at 600 nm ($OD_{600}$) of 0.4 (measured in a 96 well plate using a BioTek Synergy HT plate reader). 1 mL of prepared *S. pasteurii*, *B. sphaericus* strain 21776 or strain 21787, culture was inoculated in 250 mL Pyrex bottles with 150 mL of media (either CMM+ or CMM-) (initial concentration of biomass $OD_{600}$ = ~ 0.015). After inoculation, the systems were statically incubated at 30°C for kinetic experiments.

### 2.4. Anaerobic experiments

Pilot cultures for anaerobic experiments were limited to the use of *S. pasteurii* and were grown in the same manner as for those used in aerobic experiments. However, cells were transferred into an anaerobic chamber and re-suspended in anaerobically prepared CMM-, then transferred to a serum bottle, sealed and crimped inside the anaerobic chamber. Optical density measurements for time zero were taken after the final suspension, with the same initial density as the aerobic experiments ($OD_{600}$ = ~ 0.015). The first set of experiments investigated cell growth and ureolysis under oxygen-free conditions with a range of potential terminal electron acceptors (TEAs). Experiments were run using a batch system, consisting of 100 mL of CMM- media including 10 mM $NO_3^-$, $SO_4^{2-}$, or $Fe^{3+}$ as potential TEAs and inoculated with 1 mL of *S. pasteurii* in 150 mL serum bottles. Concentrated stock solutions of each terminal electron acceptor were made in the anaerobic chamber and filter sterilized: *i)* a concentrated $NO_3^-$ solution was made using 1M $NaNO_3$; *ii)* a concentrated $SO_4^{2-}$ solution was made by combining 1M $Na_2SO_4$ and 1M $Na_2S$, $Na_2S$ was added to quench any residual oxygen and make $SO_4^{2-}$ reduction possible; and *iii)* a stock solution of Fe(III) citrate was made using 50 mM Fe(III) citrate as previously described (Gerlach et al., 2011). Appropriate amounts of each stock solution were added to the separate serum bottles containing CMM- and *S. pasteurii*. The growth survey was also conducted in CMM- without the addition of a TEA. After inoculation, the systems were statically incubated at 30°C. Abiotic control experiments, without the inclusion of *S. pasteurii*, were also performed.





Aliquots were extracted from the systems in the anaerobic chamber and monitored for pH and $OD_{600}$ during the
duration of the experiments. Aerobic control experiments were also performed with CMM- media including 10 mM
$NO_3^-$, $SO_4^{2-}$, or $Fe^{3+}$ and inoculated with 1 mL of *S. pasteurii* in 150 mL serum bottles.
The second set of experiments investigated the detailed kinetics of ureolysis and $CaCO_3$ precipitation with *S.*
*pasteurii* and CMM+ as described above, with $NO_3^-$ as the potential TEA, by monitoring pH, dissolved $Ca^{2+}$ and $NH_4^+$
concentrations. $NO_3^-$ was chosen as it demonstrated higher growth at early stages and a modest increase in pH
compared to the other TEAs from the first set of experiments. Control experiments were also performed with CMM+
without the addition of a TEA and CMM- with $NO_3^-$ as a potential TEA. Here, a stock solution of 10 M $NaNO_3$ was
made by mixing and filter sterilizing in the anaerobic chamber, and an appropriate amount was added to the CMM+
or CMM- to reach a final concentration of 1M $NO_3^-$.
**2.5. Experimental sampling and analysis**
At different time points, 3 mL of sample were aseptically extracted from the systems and measurements were
made of pH, biomass, $NH_4^+$ and $Ca^{2+}$ concentration. $NH_4^+$ concentrations were determined using the Nessler Assay
(Mitchell and Ferris, 2005). $Ca^{2+}$ concentrations were measured after appropriate dilution in trace-metal grade 5%
$HNO_3$ (Fisher Scientific) using an Agilent 7500ce Inductively Coupled Plasma Mass Spectrometer (ICP-MS).
Bacterial biomass was determined using three methods: plate counts, $OD_{600}$ and protein assays. $OD_{600}$ was used as a
growth indicator in experiments carried out in the absence of $Ca^{2+}$ (see SI section 1.2). Transmission Electron
Microscopy (TEM) images were taken using a LEO 912AB TEM and photographed with a Proscan 2048x2048 CCD
camera from a batch culture in CMM+ inoculated with *S. pasteurii*. At the point of crystal formation (after
approximately 2.5 h), a mixture of $CaCO_3$ crystals and cells were extracted from the system. Separate samples of *S.*
*pasteurii* grown in the absence of $Ca^{2+}$ were also collected and imaged. Further details are given in the SI section 1.3.
The PHREEQC (version 2) speciation-solubility geochemical model (Parkhust and Appelo, 1999) was used to
calculate solution speciation and carbonate mineral saturation. Simulation was performed using calcite -$CaCO_3$- as
the only precipitate as it was identified by XRD (data not shown) as the calcium carbonate polymorph present in the
systems. The MINTEQ database was used for all calculations and the thermodynamic constants for urea (Stokes,
1967) were added (more information is provided in SI1.4).
**2.6. Kinetics of ureolysis and $CaCO_3$ precipitation**
The rate coefficient for ureolysis was determined by integrating the following first order differential equation
assuming constant biomass concentrations during the period of urea hydrolysis (Ferris et al., 2004; Mitchell and Ferris,

30   2005):

$$\frac{d[Urea]}{dt} = -k_{urea}[Urea][X] \quad (9)$$
to get:
$$[Urea]_t = [Urea]_o e^{-k_{urea}Xt} \quad (10)$$





where $k_{urea}$ is the first order rate coefficient for ureolysis, $t$ is the time, and $X$ is the concentration of biomass (SI,
section SI2.1). The change in urea concentration was determined according to Equations 3 and 4 (Equation 11).
$$\Delta[Urea] = -0.5 * \Delta[NH_4^+] \quad (11)$$
Ureolysis rates were calculated in two ways. Firstly, it was assumed that the reaction is zero order with respect to
biomass ($X = 0$), as performed in other studies of ureolysis kinetics (Cuthbert et al., 2012;Ferris et al., 2004;Mitchell
and Ferris, 2005, 2006;Schultz et al., 2011;Tobler et al., 2011), thus $k_{urea}$ rates were not normalized to biomass.
Secondly, $k_{urea}$ rates were normalized to the biomass concentration at the onset of precipitation (normalized to the
absorbance reading of initial biomass, $OD_{600}$, and CFU mL$^{-1}$; SI section 2.2), which was equivalent to the initial
biomass in each system ($X = X_0$). This is an appropriate choice of model, since the biomass analysis indicated that the
cell density was constant for the duration of $CaCO_3$ precipitation and was equivalent to the initial biomass in the
systems, as presented in the results section. The kinetic parameters obtained in this study ($k_{urea}$ normalized to biomass)
were compared to other parameters previously published (Ferris et al., 2004;Fujita et al., 2000;Stocks-Fischer et al.,
1999;Tobler et al., 2011). The media used in the different studies were similar to those used in this study, all with 25
mM of $Ca^{2+}$, 333 mM urea and including nutrient broth-based growth media, apart from: *i)* Ferris et al. (2004) who
used a dilute artificial groundwater (non-growth medium) with $Ca^{2+}$ and urea concentrations of 1.75 mM and 6 mM,
respectively, and *ii)* Tobler et al. (2011) who used different $Ca^{2+}$ concentrations varying from 50 to 500 mM and urea
concentrations between 250 and 500 mM.
The precipitation of $CaCO_3$ from the system is dependent on the saturation state of the system, as well as the
growth mechanism of $CaCO_3$ (Ferris et al., 2004;Mitchell and Ferris, 2005;Teng et al., 2000). The literature is
ambiguous on defining a set rate expression for $CaCO_3$ precipitation, so a non-affinity-based first order rate law was
applied to these studies for both its simplicity and the fact that it seems to describe the data well, assuming that for
every mole of $Ca^{2+}$ removed from solution one mole of $CaCO_3$ formed (Teng et al., 2000):
$$\frac{d[Ca^{2+}]}{dt} = -k_{precip}[Ca^{2+}] \quad (12)$$
Integration of the above equation yields:
$$[Ca^{2+}]_t = [Ca^{2+}]_o e^{-k_{precip}t} \quad (13)$$
where $k_{precip}$ is the first order rate coefficient for $CaCO_3$ precipitation. Rate constants were found using a non-linear
regression method utilizing the Solver function in Microsoft Excel. Due to the significant lag phase, the data used for
analysis excluded onset time before ureolysis and $CaCO_3$ precipitation occurred, as previously documented (Tobler et
al., 2011).



**3. Results and Discussion**
**3.1. Aerobic experiments**
**3.1.1 Solution chemistry**
Aerobic experiments with CMM+ medium inoculated with *S. pasteurii, B. sphaericus* 21776 and *B. sphaericus*
21787 showed an increase in pH (Table 1) and $NH_4^+$ (displayed as a stoichiometrically equivalent decrease in urea
concentrations, Figure 1) over time. These results support observations from previous studies confirming the ureolytic
capabilities of *S. pasteurii* (Ferris et al., 2004;Fujita et al., 2000;Warren et al., 2001) and *B. sphaericus* species (Dick
et al., 2006;Hammes et al., 2003a). Differences in the rate of pH change and the amount of urea hydrolyzed between
the different bacterial species suggest differences in their ureolytic activity. After 30 h, 58-82% and 72-80% of the
available urea was hydrolyzed by *S. pasteurii* and *B. sphaericus* 21776, respectively. *B. sphaericus* 21787 exhibited
little utilization of urea (12-15% hydrolyzed) accompanied by only more slight increase in pH values (~ pH 8.7 by 24
$\pm$ 3 h) compared to the other bacterial strains [~ pH 9.3 by 24 $\pm$ 3 h, consistent with buffering by $NH_4^+ \leftrightarrow NH_3 + H^+$
which has a $pK_a$ value of 9.3 at 30°C (Mitchell and Ferris, 2005)]. Control experiments, inoculated with the non-
ureolytic organism *B. subtilis* and sterile controls, did not exhibit significant changes in pH (Table 1) or urea
concentrations (data not shown). Geochemical modelling suggested that no $CaCO_3$ precipitation should occur in the
absence of ureolysis and that approximately 25.5 mM of urea would have had to be hydrolyzed to achieve
supersaturation and for $CaCO_3$ precipitation to commence (see SI1.4).
In all the experiments containing ureolytic bacteria, $Ca^{2+}$ concentration decreased to ~ 5% of the initial $Ca^{2+}$
concentration in the liquid medium after approximately 30 h (Figure 1). The decrease of $Ca^{2+}$ concentrations suggests
the precipitation of $CaCO_3$, which was identified by XRD (data not shown) as the polymorph calcite. The onset of
$CaCO_3$ precipitation occurred shortly after the start of the experiment (~3-4 h) in the *S. pasteurii* and *B. sphaericus*
21776 experiments, whereas the onset of precipitation was delayed (~ 9 h) for *B. sphaericus* 21787. This supports
differences in the rate of urea hydrolysis, and thus the time at which $CaCO_3$ saturation was exceeded (Equations 2, 4,
5 and 7). Differences in ureolysis and $CaCO_3$ precipitation rates can be attributed to differences in the specific ureolytic
activities of the organisms or the number of ureolytically active cells. It has been suggested that the inherent variation
between different organisms' ability to produce the urease enzyme should affect the rates of ureolysis and associated
rates of $CaCO_3$ precipitation (Anbu et al., 2016), as well as the characteristics of the resulting $CaCO_3$ precipitates
(Hammes et al., 2003a).
**3.1.2 Aerobic bacterial growth and ureolytic activity**
Changes in biomass, measured as protein and colony forming units (CFUs), were observed during ureolysis in both
aerobic CMM+ and CMM- experiments (Figure 2). CFU and protein concentrations exhibited similar trends for *S.*
*pasteurii* and *B. sphaericus* 21776 where in CMM- experiments, CFUs and protein increased over time asymptotically
(Figure 2). In CMM+ experiments, CFUs and protein concentrations seemed to slightly decrease or remain quasi-
constant while $CaCO_3$ precipitation occurred (< 10 h), followed by an increase in CFUs and protein once $Ca^{2+}$ had
been depleted (Figures 1 and 2). Decrease of biomass growth during $CaCO_3$ precipitation has been suggested to occur



due to the encasement of bacteria within the CaCO₃ precipitates (Tobler et al. 2011). Encasement of *S. pasteurii* cells
in CaCO₃ minerals has been reported (Cuthbert et al., 2012;Ebigbo et al., 2012;Schultz et al., 2011;Stocks-Fischer et
al., 1999) and cell indentations in CaCO₃ precipitates have been observed (Mitchell and Ferris, 2005). The recovery
(*i.e.* re-growth) of biomass after CaCO₃ precipitation suggests that ureolysis-induced CaCO₃ precipitation does not
have to be a lethal event for the population as a whole, and that net cell growth can resume after CaCO₃ precipitation
ceases (Figure 2). For both *B. sphaericus* strains, in contrast to *S. pasteurii,* CFUs were higher in the CMM+
experiments, despite protein concentrations being lower in the CMM+ experiments. This might suggest that cell
mortality of *B. sphaericus* strains is increased in the calcium-free experiments, which may reflect lower tolerance to
the higher pH values generally observed in the calcium-free experiments (Table 1, Figure 2).
TEM images and electron energy loss spectroscopy (EELS) of material collected on 0.2 μm pore size filters
from the CMM+ *S. pasteurii* systems (Figure 3A-C) confirm that some cells are surrounded by a layer of calcium-
containing precipitates. Figure 3D shows *S. pasteurii* grown in CMM- for comparison. The data suggest that cells are
removed from suspension and potentially inactivated by CaCO₃ encasement, either in large crystals (Mitchell and
Ferris, 2005) or by a thin coating (Figure 3A-C).
Quasi-constant biomass concentrations during CaCO₃ precipitation (Figure 2) suggest that cell growth might
not have to be considered in kinetic descriptions of bacterially induced CaCO₃ precipitation. However, it is unclear
whether *(i)* CaCO₃-encased cells are ureolytically active or *(ii)* CaCO₃ precipitates surrounding the cells effectively
act as a barrier to urea reaching the cell or to NH₃, OH⁻, or NH₄⁺ formed by the hydrolysis of urea from diffusing
through the CaCO₃ to the bulk solution. Therefore, a theoretical analysis of urea diffusion in CaCO₃ was performed.
The diffusion of oxygen in CaCO₃ at high temperatures has been documented (Farver, 1994), but, to the best of our
knowledge, information of urea diffusion in CaCO₃ at 30°C has not been reported. A number of assumptions were
made for the estimates in this study: *(1)* since urea has a lower diffusion coefficient than oxygen in aqueous solutions
at 25°C (Stewart, 2003), it was assumed that this will hold true at other temperatures and through other substances,
like CaCO₃; *(2)* since the diffusion coefficient of oxygen through CaCO₃ at 400°C and 100 MPa is 2.66 x 10⁻²² m² s⁻¹
(Farver, 1994), and diffusion coefficients generally increase with increasing temperature and pressure, it can be
assumed that the diffusion coefficient of urea in CaCO₃ at atmospheric pressure and 30°C is smaller than 2.66 x 10⁻²²
m² s⁻¹; *(3)* assuming that the geometry of the CaCO₃ is a uniformly thick slab with a thickness of approximately 200
nm, as determined from the TEM images (Figure 3B), the time it will take to reach 5% of the bulk urea concentration
can be calculated using the relation presented by (Carslaw and Jaeger, 1959):
$$t_5 = 0.1 \frac{L^2}{D_e} \quad (13)$$

where $L$ is the slab thickness, $D_e$ is the (estimated) effective diffusion coefficient in CaCO₃, and $t_5$ is the amount of
time it will take to reach 5% of the bulk concentration. Using the above assumptions, it would take at least 175 days
for 5% of the urea to diffuse through the CaCO₃ surrounding the cells. Because CaCO₃ precipitation takes place over
the course of approximately one day, it can safely be assumed that even if the encased cells are still alive, urea is not
able to diffuse through the CaCO₃ fast enough for them to hydrolyze significant amounts and contribute to the increase
in solution alkalinity. Therefore, it is argued that, at least in the systems described here, cell growth does not have to



be considered in kinetic expressions describing ureolysis during CaCO$_3$ precipitation. Instead, the biomass at the onset
of precipitation, which is equivalent to the initial biomass in the system (Figure 2), can be used to normalize the
observed ureolysis rates to biomass concentration.
While a physical association between cells and CaCO$_3$ precipitates is evident, precipitation is likely to occur
from a combination of *(i)* homogeneous nucleation in the bulk solution, in alkaline microenvironments around
bacterial cells (Schultze-Lam et al., 1996;Stocks-Fischer et al., 1999), and *(ii)* heterogeneous nucleation on nascent
crystals, bottle walls and the bacterial cell surfaces (Rodriguez-Navarro et al., 2012).
**3.1.3 Kinetics of ureolysis and CaCO$_3$ precipitation**
Kinetic analyses were performed on the individual CMM+ and CMM- experimental data for all bacterial strains
(Figure SI2.1). A summary of the parameters estimated is shown in Table 2 (for detailed results on individual
experiments see Table SI2.1). $k_{urea}$ values for the bacterial species varied according to the presence (CMM+) or
absence (CMM-) of Ca$^{2+}$ in the medium. *S. pasteurii* and *B. sphaericus* 21776 exhibited similar $k_{urea}$ values in CMM-
and CMM+ systems with $k_{urea}$ values being between 1.6 and 2.5 times higher in the absence of Ca$^{2+}$ (*S. pasteurii*:
$k_{urea,CMM+} = 0.07 \pm 0.02$ h$^{-1}$, $k_{urea,CMM-} = 0.19 \pm 0.10$ h$^{-1}$; *B. sphaericus* 21776: $k_{urea,CMM+} = 0.10 \pm 0.03$ h$^{-1}$, $k_{urea,CMM-} =$
$0.16 \pm 0.05$ h$^{-1}$). This is likely due to the encasement of cells by CaCO$_3$ and their inactivation in CMM+ experiments.
Some data points were excluded for *S. pasteurii* CMM- $k_{urea}$ calculations because of an estimated increase in urea
concentration (based on a decrease in NH$_4^+$ concentration) likely due to significant volatilization of NH$_4^+ \leftrightarrow$ NH$_3$ +
H$^+$ that can occur at pH > 9 (at 34 h; open marker) (Figure SI2.1B). *B. sphaericus* 21787 exhibited low $k_{urea}$ values in
both CMM+ ($k_{urea} = 0.02$ h$^{-1}$) and CMM- ($k_{urea} = 0.05$ h$^{-1}$). From triplicate experiments, only one experiment showed
values that could be used for kinetic analysis (Figure SI2.1), and some outlying data points were not used for the
kinetic calculations, hence no standard deviations can be provided (Table 2). Thus, rate coefficients obtained for *B.*
*sphaericus* 21787 are not statistically valid but were estimated for the purpose of comparison to the other studied
bacterial strains. The obtained results suggest that *B. sphaericus* 21787 exhibits limited ureolytic capabilities under
the experimental conditions used in this study.
A lag time before the onset of ureolysis was observed for all bacterial strains. *S. pasteurii* exhibited a lag time
of ~4 h in both CMM+ and CMM-, and *B. sphaericus* 21776 ~5 h in CMM+ and ~3 h in CMM-. Connolly et al. (2013)
observed a lag time to the onset of ureolysis of ~15 h for *S. pasteurii*, ~6 h for *Pseudomonas aeruginosa* MJK1, and
~4 h for *Escherichia coli* MJK2, when cultivated with 0.16 mM urea in similarly composed CMM-. In the present
study, $k_{urea}$ values normalized to the initial biomass concentration showed similar trends for *S. pasteurii* and *B.*
*sphaericus* 21776, suggesting similar cell specific urease activity between the strains. The standard deviations of $k_{urea}$,
initial biomass and lag time were small between replicate experiments with *S. pasteurii* and *B. sphaericus* 21776, and
R$^2$ values for the fit to Equation 10 were greater than 0.9 (Table SI2.1).
The kinetic parameters obtained in this study were compared to other parameters previously published (Tobler
et al., 2011; Ferris et al., 2004; Fujita et al., 2000; Stocks-Fischer et al., 1999). $k_{urea}$ values of *S. pasteurii* obtained in
this study as well as in previous publications were standardized to the initial cell concentrations (Table 3). Values of
$k_{urea}$ were higher in the present study for both *S. pasteurii* and *B. sphaericus* 21776 ($k_{urea} = 0.07$ h$^{-1}$ and 0.11 h$^{-1}$,





respectively) than those from *S. pasteurii* in other studies ($k_{urea}$ = 0.008 to 0.028 $h^{-1}$), except for the results obtained
by Tobler et al. (2011) ($k_{urea}$= 0.13 to 2.29 $h^{-1}$). This was also apparent once normalized to biomass (this study, *B.*
*sphaericus* 21776 $k_{urea}$ = 7.64 $OD_{600}^{-1}$ $h^{-1}$ and *S. pasteurii* 5.28 $OD_{600}^{-1}$ $h^{-1}$, compared to $k_{urea}$ = 0.11 to 2.80 $OD_{600}^{-1}$ $h^{-}$
$^{1}$ in previous studies, and 4.33 to 32.71 $OD_{600}^{-1}$ $h^{-1}$ for Tobler et al. 2011). The generally higher $k_{urea}$ values in this
study appear to reflect the higher temperature (30°C) used compared to the previous studies, which ranged from 20-
25°C. Higher temperatures generally increase reaction rates where chemical reactions are advanced through a transient
activated complex (Stumm and Morgan, 1996). In urease, the transitional state involves coordination of urea and water
at the active catalytic site of the enzyme (Jabri et al., 1995). Formation of such an activated complex tends to impart
a greater temperature dependency on the absolute reaction rate than would be encountered if the reactions were
mediated solely by collisions arising from molecular diffusion (Ferris et al., 2004;Mitchell and Ferris, 2005).

11       The biomass concentration-normalized $k_{urea}$ values from Fujita et al. (2000) are much lower. This could be due

to the highest biomass concentrations ($OD_{600}$ = 0.072) used in these studies; very high biomass concentrations could
shift the primary kinetic dependency from being catalyst (i.e. enzyme limited) to substrate limited. However, this
appears to be opposite to what Tobler et al. (2011) reported indicating $k_{urea}$ increased with increasing inoculum density.
The rate constants obtained with the three organisms used in the present study are similar to the range of values
measured in deeper vadose zone mineral subsoils which were between 0.00375 $h^{-1}$ to 0.07 $h^{-1}$ (Swensen and Bakken,
1998), suggesting natural levels of ureolytic bacterial activity were reasonably approximated in the aerobic
experiments.

19       On average, *B. sphaericus* 21776 had the highest $k_{precip}$ (0.60 ± 0.34 $h^{-1}$), although considering its high standard

deviation, $k_{precip}$ for *S. pasteurii* is not significantly different ($k_{precip}$ = 0.25 ± 0.02 $h^{-1}$). $R^2$ values of the fit to Equation
12 were relatively high (0.84 − 0.93) for *B. sphaericus* 21776 and *S. pasteurii* (Table 2). The lag time for $CaCO_3$
precipitation was 3.3 h for all bacterial strains, which reflects the similar $k_{urea}$ values, and thus the time it took to reach
$CaCO_3$ saturation and induce precipitation. Tobler et al. (2011) observed similar lag times until the onset of $CaCO_3$
precipitation (2-3 h) in aerobic experiments (artificial groundwater with no nutrients added, 250-500 mM urea and 50-
00 mM Ca) with *S. pasteurii*. First order rate constants for $CaCO_3$ precipitation observed here were also higher (0.21
$h^{-1}$<$k_{precip}$<0.60 $h^{-1}$) compared to other studies (0.01 $h^{-1}$<$k_{precip}$<0.11 $h^{-1}$) (Table 3). This is likely associated with the
greater $k_{urea}$ values observed in this study than in previous studies. Temperature is unlikely to account for this variation
given the modest decrease in calcite solubility (~ 27 %) that occurs between 20°C and 30°C (Miller, 1952;Stumm and
Morgan, 1996). Overall, $k_{urea}$ values are lower than $k_{precip}$ values in this and previous studies, with the exception of
Ferris et al. (2004), indicating urea hydrolysis is the rate limiting step during ureolysis-induced $CaCO_3$ precipitation
and that $CaCO_3$ precipitation rates are rapid once the critical supersaturation is exceeded (Mitchell and Ferris, 2006).
**3.2. Anaerobic experiments**
**3.2.1 Anaerobic ureolysis and bacterial growth**

34       Given the potential for anaerobic conditions in subsurface environments, screening experiments were

performed to assess the capability of *S. pasteurii* to grow and/or increase pH in CMM- in the presence of $NO_3^-$, $SO_4^{2-}$
, and $Fe^{3+}$ as potential TEAs. There are contradicting reports in the literature regarding *S. pasteurii*'s ability to grow





and hydrolyze urea in the absence of oxygen; some studies suggest that the anoxic environment does not hinder urease
activity (Mortensen et al., 2011;Tobler et al., 2011), whereas other studies report limited microbial growth and poor
ureolysis (Martin et al., 2012).

4  In this study, pH increased (pH > 9.0) in all experiments with and without potential TEAs added (Figure 4),

suggesting ureolysis by *S. pasteurii* took place in the absence of oxygen. The pH of the abiotic controls did not exceed
pH 7, except for the medium containing $SO_4^{2-}$ as the added TEA, which had an initial pH of 7.7, which remained
constant throughout the experiments (Figure 4C). Growth in the CMM- anaerobic experiments quantified by $OD_{600}$
absorbance was lower than that observed in aerobic experiments regardless of the presence or absence of TEAs (Figure
4). Although some growth might have occurred during the initial period of the experiments (increased absorbance by
~20 h), the lack of sustained growth over time suggests the inability of *S. pasteurii* to grow under anaerobic conditions.
These findings are in agreement with those by Martin et al. (2012), who observed limited cell growth in the absence
of oxygen. In the present study, once oxygen was allowed to diffuse into the systems (at 120 h), optical density in all
inoculated systems increased, indicating that even though no significant growth was observed in the absence of
oxygen, the bacteria were still viable after 120 h of oxygen depletion and can be resuscitated.

15  Sustained growth of *S. pasteurii* in the absence of oxygen does not appear to be feasible which might limit the

potential use of *S. pasteurii* for inducing $CaCO_3$ precipitation in the subsurface to only short-term purposes. However,
the potential regrowth of microbes, even after prolonged periods of exposure to oxygen-free conditions, suggests that
*S. pasteurii* could be resuscitated and re-stimulated through the injection of oxygenated fluids, which could enable
bacterial growth and thus ureolytic activity over longer periods of time.
**3.2.2 Kinetics of anaerobic ureolysis and $CaCO_3$ precipitation**

21  After the screening experiments with different TEAs, studies were performed to determine the kinetics of

ureolysis and $CaCO_3$ precipitation in the absence of oxygen. For these in-depth studies, only $NO_3^-$ was used as a
potential TEA since slight growth and a modest increase in pH were observed in the anaerobic ureolysis and bacterial
growth studies (Figure 4A). Average change in urea and $Ca^{2+}$ concentrations for CMM+ and CMM- with $NO_3^-$ (Figure
5), and CMM+ without TEA added are presented (Figure 5). Urea was hydrolyzed under all experimental conditions
and $CaCO_3$ precipitation was observed in the presence of $Ca^{2+}$ (Figure 5). However, ureolysis and $CaCO_3$ precipitation
did not occur in all replicates for each experiment, which accounts for the high standard deviations.

28  Rate coefficients were estimated as previously described for aerobic experiments (Figure SI2.2) and a summary

of the results is presented in Table 2 (for detailed results for individual experiments see Table SI2.2). Data points
which preceded the onset and completion of ureolysis and calcite precipitation were excluded (Figure SI2.2), $k_{urea}$
seems to be lowest in CMM+ with $NO_3^-$ ($k_{urea}$ = 0.04 ± 0.01 h$^{-1}$), followed by CMM- with $NO_3^-$ ($k_{urea}$ = 0.07 ± 0.01 h$^{-1}$
$^{1}$) and CMM+ without TEA ($k_{urea}$ = 0.08 h$^{-1}$). The same relative rates were apparent when $k_{urea}$ values were normalized
to biomass (Table 2). The presence of $NO_3^-$ appears to have slightly decreased ureolysis rates. While the reasons for
this possible decrease in the ureolytic activity due to the presence of $NO_3^-$ are unclear, it could be perceivable that the
ureolytic activity is down-regulated in the presence of the alternative nitrogen source $NO_3^-$. Longer lag times to the
onset of ureolysis (ranging from 6.5 to 10 h) were observed in the anaerobic experiments relative to the aerobic ones.





Moreover, comparing the experiments containing $NO_3^-$, the presence of $Ca^{2+}$ in the medium results in a lower $k_{urea}$
value (Table 2). This could be due to the encasement of cells by $CaCO_3$.
The onset of $CaCO_3$ precipitation occurred after approximately 6 h, which was twice the lag time observed for
precipitation under aerobic conditions (Table 2); this was expected as slower ureolysis was observed in the anaerobic
experiments (Table 2). Rate constants ($k_{precip}$) for anaerobic $CaCO_3$ precipitation were not statistically different for
CMM+ medium with $NO_3^-$ (0.36 h$^{-1}$ ± 0.22) and CMM+ medium without TEA (0.19 h$^{-1}$ ± 0.05). Tobler et al. (2011)
reported a similar $k_{urea}$ value (0.09 h$^{-1}$) for experiments with *S. pasteurii* under anoxic conditions in natural
groundwater containing comparable concentrations of urea and $Ca^{2+}$ (250 mM and 50 mM, respectively); $k_{precip}$ was
not reported due to a poor fit of the data with first order kinetics. Experiments containing only indigenous bacteria
exhibited far lower rates of ureolysis ($k_{urea}$ = 0.0016 h$^{-1}$) and $CaCO_3$ precipitation ($k_{precip}$ = 0.009 h$^{-1}$) than observed in
the present study. Differences in ureolysis and $CaCO_3$ precipitation rates between this study and Tobler et al.'s (2011)
study is likely due to the low initial biomass of the indigenous ureolytic population in the natural groundwater.
Comparison of kinetics from aerobic and anaerobic experiments in the present study demonstrates that rates
are on the same order of magnitude (Table 2). In CMM+, urea hydrolysis rates for *S. pasteurii* under aerobic and
anaerobic conditions (with or without TEA) were not significantly different ($P_{value}$=0.274), even when normalized to
initial biomass concentrations ($OD_{600}$ or CFU mL$^{-1}$). pH increases in the screening experiments suggest anaerobic
ureolysis occurred at the same rate as under aerobic conditions (Figure 4). Similarly, $k_{precip}$ in aerobic and anaerobic
experiments are comparable (0.19 h$^{-1}$ and 0.25 h$^{-1}$, respectively). This suggests that oxygen-free environments do not
significantly impact the rate of ureolysis or $CaCO_3$ precipitation initially, but that anaerobic growth cannot be
conclusively demonstrated under the conditions of this present study. This supports observations by Tobler et al.
(2011), who reported similar rates of ureolysis under both oxic and anoxic conditions when amending natural
groundwater with *S. pasteurii* ($k_{urea}$ 0.10 h$^{-1}$ in oxic conditions, $k_{urea}$ 0.09 h$^{-1}$ in anoxic conditions with 50 mM $Ca^{2+}$
and 250 mM urea, cf. Table 3). Martin et al. (2012) also observed ureolytic activity by *S. pasteurii* under anoxic
conditions but to a lesser extent compared to the extensive activity reported by Tobler et al. (2011), however rate
constants were not reported. The current study therefore suggests ureolytic activity observed under anoxic conditions
corresponds to the urease already present in the cells as suggested by Martin et al (2012). *S. pasteurii* could therefore
potentially be used for $CaCO_3$-induced precipitation in the subsurface in the short-term, and the bacterial growth could
be stimulated through multiple injection of bacterial cells or oxygenated medium to re-enable ureolytic activity and
thus $CaCO_3$ precipitation. This is supported by our calcite precipitation rate constants under anaerobic conditions, the
first to be reported for a ureolytic strain, which are comparable to aerobic rate constants, suggesting anaerobic
conditions will not significantly inhibit $CaCO_3$-induced precipitation in the subsurface.
**4. Conclusions**
All three ureolytic strains studied, *S. pasteurii* as well as *B. sphaericus* strains 21776 and 21787, were capable
of inducing $CaCO_3$ precipitation under aerobic conditions. Data obtained in this study suggest that rates of ureolysis
and ureolysis-induced $CaCO_3$ precipitation are affected by differences in the ureolytic species. This information should
be considered in subsurface engineering strategies utilizing microbial amendment or stimulation. Specifically, rates



of ureolysis and CaCO₃ precipitation were comparable for *S. pasteurii* and *B. sphaericus* 21776. Although *B.*
*sphaericus* 21787 showed poor ureolysis, some CaCO₃ precipitation was observed. When rate coefficients were
normalized to cell numbers, *B. sphaericus* 21776 had the highest rate of ureolysis and calcite precipitation per cell
compared to *S. pasteurii* and *B. sphaericus* 21787, indicating that it may have a higher cell-specific ureolytic activity
under the conditions studied here. *B. sphaericus* 21776 may therefore be a candidate species for subsurface
augmentation if maximizing rates of ureolysis and precipitation is desirable, although in our experiments we were not
able to generate cell concentrations as high as in *S. pasteurii* cultures, so optimization of growth media may be
required.

9        *S. pasteurii* was capable of ureolysis in anaerobic environments with and without the addition of potential

electron acceptors, however, sustained growth of *S. pasteurii* over time in the absence of oxygen did not appear to be
possible. Comparison of kinetics from aerobic and anaerobic experiments demonstrates that rates are on the same
order of magnitude suggesting that oxygen-free environments do not significantly impact the initial rate of ureolysis
or CaCO₃ precipitation. Apparent rate coefficients for ureolysis were reduced in CMM+ relative to CMM-. The limited
increase in cell biomass during the period of CaCO₃ precipitation and TEM images reveal this may be due to the
encasement and inactivation of cells. However, populations can recover and re-grow once CaCO₃ precipitation has
ceased. Therefore, ureolysis-induced CaCO₃ precipitation is likely to be efficient in aerobic and anaerobic subsurface
systems. However, our data and other recent studies under flow conditions demonstrate that if only one injection of
microbes is to occur but longer term ureolysis is desired in subsurface applications, resuscitation and regrowth of
microbes, e.g. through the injection of growth media, will be necessary since CaCO₃ precipitation greatly inhibits cell
growth (Cuthbert et al., 2012;Phillips et al., 2013b).
**Author contribution**
A.C.M., E.J.E.-O., S.L.B. and R.G. wrote the manuscript. R.G., A.C.M., S.P., A.P. and A.B.C. designed
experiments. S.L.B. performed experiments.

**Competing interests**

The authors declare that they have no conflict of interest.

**Acknowledgements**
Funding was provided from the US Department of Energy (DOE) Zero Emissions Research Technology Center
(ZERT), Award No. DE-FC26-04NT42262, DOE EPSCoR Award No. DE-FG02-08ER46527, DOE Office of
Science, Subsurface Biogeochemical Research (SBR) Program, Contract No. DE-FG-02-09ER64758, DOE NETL
Contract No. DE-FE0004478 and DE-FE0009599, European Union Marie-Curie Reintegration Grant, Award No.
277005 (CO₂TRAP), National Science Foundation Award No. DMS-0934696, and a Sêr Cymru National Research
Network for Low Carbon, Energy and the Environment Grant from the Welsh Government and Higher Education
Funding Council for Wales (A.C.M). Support for the Environmental and Biofilm Mass Spectrometry Facility through
DURIP, Contract Number: W911NF0510255 and the MSU Thermal Biology Institute from the NASA Exobiology



1  Program Project NAG5-8807 is acknowledged. TEM images were kindly taken by Susan Brumfield at Montana State

2  University in the Plant Science and Plant Pathology Laboratory.





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





**Figures**
**Figure 1.** Changes in urea (▲) and dissolved calcium (●) concentrations during ureolysis over time in calcium-
inclusive aerobic experiments for (A) *S. pasteurii*, (B) *B. sphaericus* 21776, and (C) *B. sphaericus* 21787. Data points
are the averages of triplicate experiments; vertical error bars represent the standard deviations; horizontal error bars
indicate standard deviation of the sampling times; error bars are smaller than markers if not visible.
**Figure 2.** Change in protein concentrations and CFU mL$^{-1}$ over time for calcium inclusive (solid markers) and calcium
exclusive (open markers); aerobic medium, (A) *S. pasteurii*, (B) *B. sphaericus* 21776 and (C) *B. sphaericus* 21787.
Data points are the average of triplicate experiments; vertical error bars represent the standard deviation of triplicate
experiments; horizontal error bars indicate standard deviation of the sampling times.
**Figure 3.** Transmission electron microscopy images of *S. pasteurii* cells in calcium-inclusive (A-C) and calcium-
exclusive (D) media. Arrows indicate a cross-section through a Ca-rich layer, approx. 280 nm thick.
**Figure 4.** Changes in pH and OD$_{600}$ over time in anaerobic (■) and aerobic (●) calcium-exclusive medium for *S.*
*pasteurii* with (A) NO$_3^-$, (B) Fe$^{3+}$, and (C) SO$_4^{2-}$ as terminal electron acceptors; experiments without added terminal
electron acceptor are shown in (D). Abiotic controls (▲) are also shown. Open markers indicate values for timepoints
after the bottles were opened to the environment allowing oxygen to enter the system.
**Figure 5.** Changes in urea (triangle markers) and dissolved calcium (cirlce markers) concentrations over time in
anaerobic experiments with *S. pasteurii* in calcium inclusive medium with NO$_3^-$ (black solid markers), calcium
exclusive medium with NO$_3^-$ (open markers) and calcium inclusive medium without added terminal electron acceptor
(grey solid markers). Data points are the averages of triplicate experiments; vertical error bars represent the standard
deviations of measurements; horizontal error bars indicate standard deviation of the sampling times; error bars are
smaller than markers if not visible.



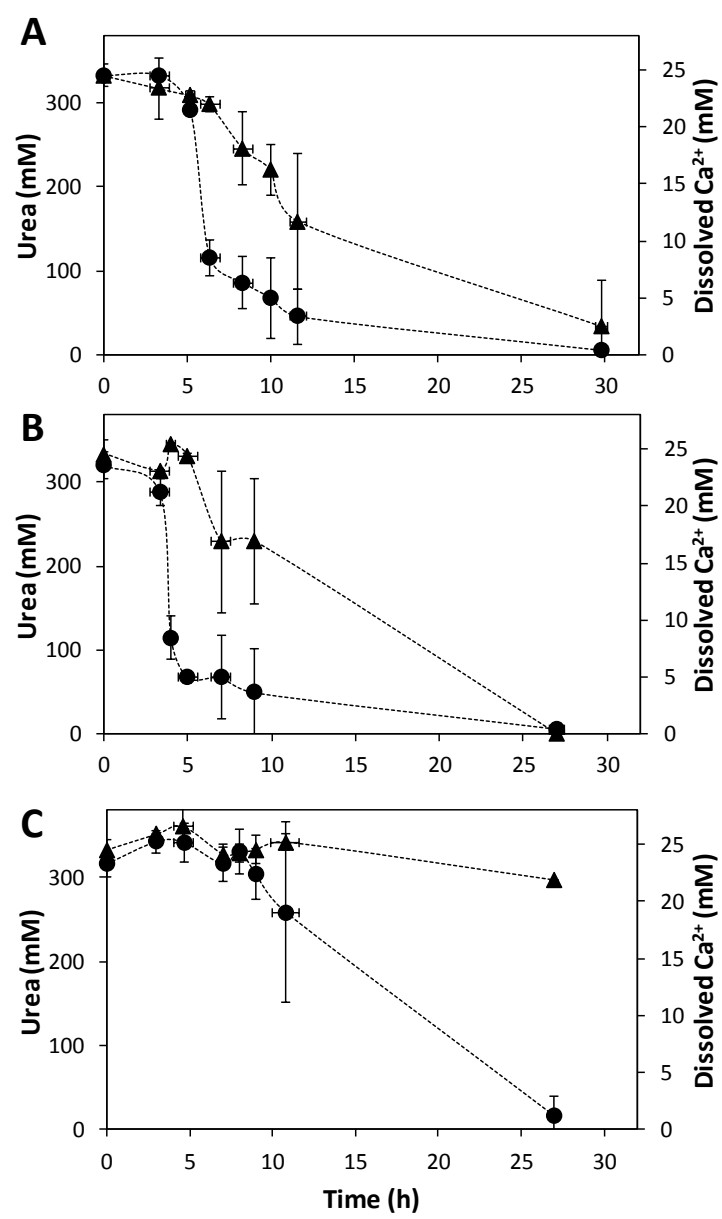

2      **Figure 1.**





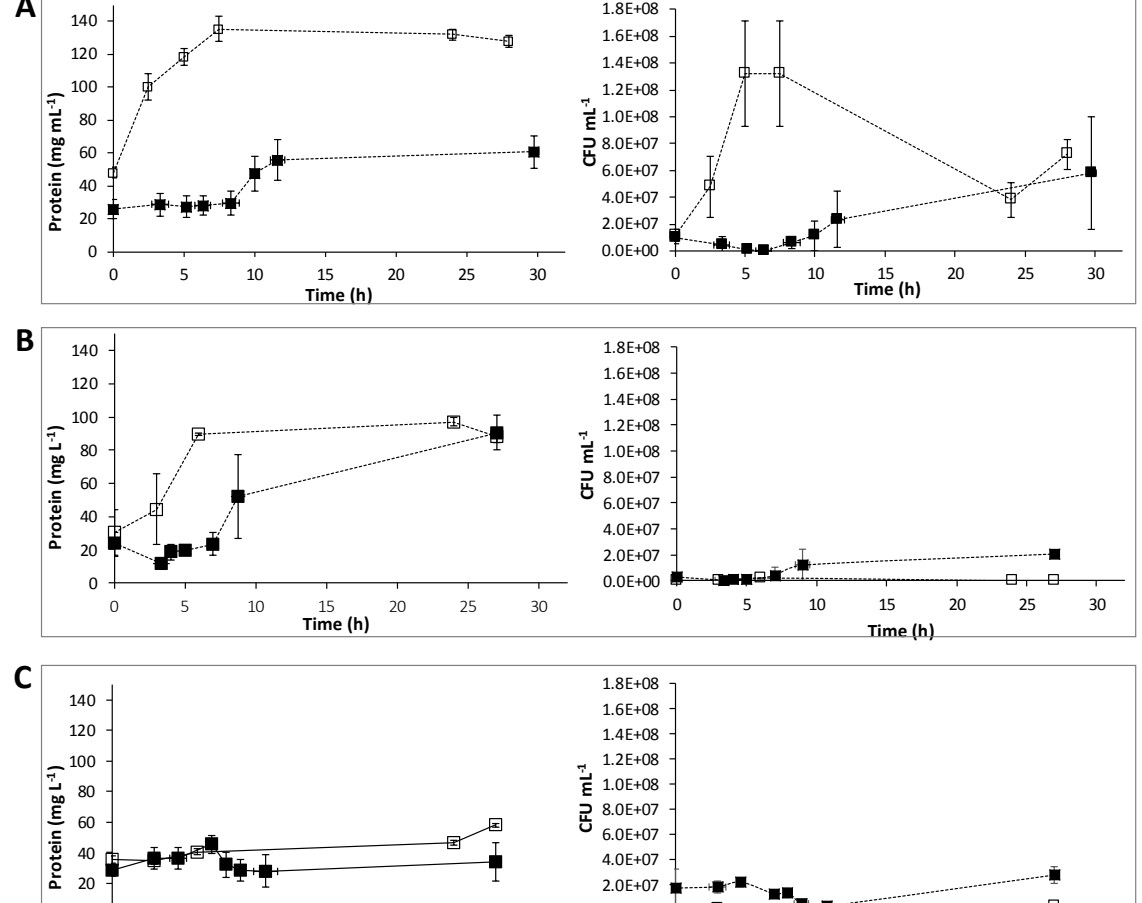

4    **Figure 2.**



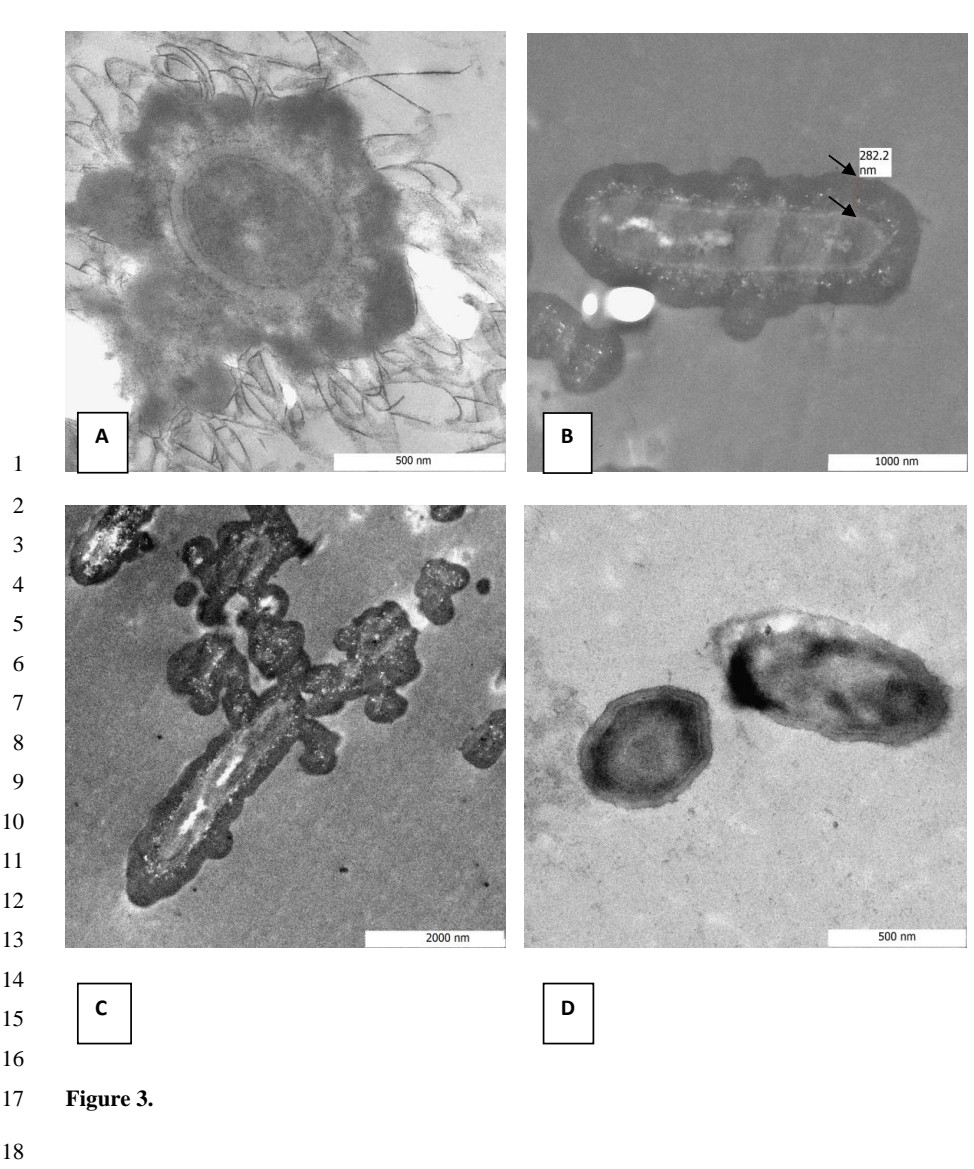

**Figure 3.**



**Figure 4.**



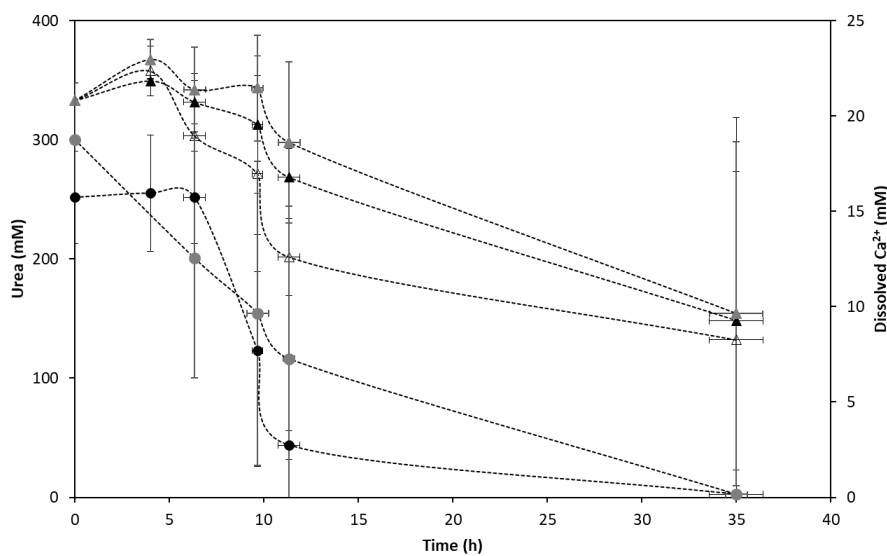

2 **Figure 5.**



**Tables**
**Table 1.** Change in pH in aerobic calcium-inclusive (CMM+) and calcium-exclusive experiments (CMM-). Results
are averages from triplicate experiments unless stated otherwise. Data for hour 0 was taken immediately after
inoculation.

|  | Species | 0 h | 10 (±1) h | 24 (±3) h |
|---|---|---|---|---|
| *CMM+* | *S. pasteurii* | 6.66±0.06 | 8.87±0.08 | 9.33±0.02* |
|  | *B. sphaericus* 21776 | 7.24±0.30 | 8.80±0.20 | 9.23±0.09* |
|  | *B. sphaericus* 21787 | 6.87±0.15 | 8.06±0.12 | 8.70±0.26* |
|  | *B. subtilis* | 6.80§ | --- | 7.50§ |
|  | Sterile Control | 7.08±0.04 | --- | 7.31±0.05* |
|  |  |  |  |  |
| CMM- | *S. pasteurii* | 6.91±1.01* | 9.16±0.12* | 9.16§ |
|  | *B. sphaericus* 21776 | 6.85±0.91* | 9.10§ | 9.30±0.14* |
|  | *B. sphaericus* 21787 | 7.5§ | --- | 9.00§ |
|  | *B. subtilis* | 7.15* | 7.40§ | --- |
|  | Sterile Control | 6.3±0.36 | 6.56±0.35 | 6.60±0.28* |

*Data taken from only two experiments
§Data taken from only one experiment
---No data available



**Table 2.** Summary of kinetic parameters for urea hydrolysis ($k_{urea}$) and calcite precipitation ($k_{precip}$) in aerobic and anaerobic experiments in calcium-inclusive (CMM+) and calcium-exclusive (CMM-) experiments inoculated with *S. pasteurii*, *B. sphaericus* 21776 and *B. sphaericus* 21787. Anaerobic experiments were incubated with or without nitrate as the terminal electron acceptor (TEA). Results are averages from triplicate experiments unless stated otherwise.

| | Microorganism | Initial biomass ($OD_{600}$) | $k_{urea}$ ($h^{-1}$) | Lag time (h) | $k_{urea}$ normalized to: $OD_{600}$ ($OD_{600}^{-1}\,h^{-1}$) | CFU (mL $CFU^{-1}\,h^{-1}$) | $k_{precip}$ ($h^{-1}$) | Lag time (h) |
|---|---|---|---|---|---|---|---|---|
| ***Aerobic*** | | | | | | | | |
| **CMM+** | *S. pasteurii* | 0.014 ± 0.001 | 0.074 ± 0.021 | 5.0 ± 1.0 | 5.251 ± 1.273 | 3.22E-08 ± 6.54E-09 | 0.253 ± 0.021 | 3.3 ± 0.6 |
| | *B. sphaericus* 21776 | 0.014 ± 0.001 | 0.107 ± 0.038 | 4.0 ± 0.0 | 8.020 ± 3.786 | 5.30E-08 ± 3.23E-08 | 0.604 ± 0.344 | 3.3 ± 0.6 |
| | *B. sphaericus* 21787 | 0.015 ± (n/a) [§] | 0.023 ± (n/a) [§] | 3 ± (n/a) [§] | 1.526 ± (n/a) [§] | 8.52E-09 ± (n/a) [§] | 0.219 ± (n/a) [§] | 8 ± (n/a) [§] |
| **CMM-** | *S. pasteurii* | 0.017 ± 0.000 [*] | 0.192 ± 0.104 [*] | 4.0 ± 0.0 [*] | 11.227 ± 5.988 [*] | 5.88E-08 ± 2.02E-08 [*] | - | - |
| | *B. sphaericus* 21776 | 0.015 ± 0.001 [*] | 0.168 ± 0.050 [*] | 3.5 ± 0.71 [*] | 10.818 ± 2.797 [*] | 6.20E-08 ± 1.33E-08 [*] | - | - |
| | *B. sphaericus* 21787 | 0.015 ± (n/a) [§] | 0.067 ± (n/a) [§] | 6.0 ± (n/a) [§] | 3.196 ± (n/a) [§] | 1.75E-08 ± (n/a) [§] | - | - |
| ***Anaerobic*** | | | | | | | | |
| **CMM+** | *S. pasteurii*/$NO_3^-$ | 0.014 ± 0.002 [*] | 0.048 ± 0.018 [*] | 6.5 ± 0.7 [*] | 3.617 ± (1.092) [*] | 2.34E-08 ± 4.67E-09 [*] | 0.360 ± 0.223 | 6.5 ± 0.6 |
| | *S. pasteurii*/no TEA | 0.014 ± 0.002 | 0.082 ± (n/a) [§] | 10.0 ± (n/a) [§] | 6.616 ± (n/a) [§] | 4.68E-08 ± (n/a) [§] | 0.191 ± 0.050 | 3.3 ± 0.6 |
| **CMM-** | *S. pasteurii*/$NO_3^-$ | 0.014 ± 0.002 [*] | 0.071 ± 0.017 [*] | 8.5 ± 2.1 [*] | 5.278 ± 0.875 [*] | 3.45E-08 ± 2.05E-09 [*] | - | - |

§ One experiment used in analysis

* Two experiments used for kinetic analysis

n/a = No data available





**Table 3.** Summary of kinetic coefficients and initial growth conditions for aerobic calcium-inclusive experiments performed in this study and previous studies.

| Aerobic conditions | This study | This study | This study | Stocks-Fischer et al. (1999) | Fujita et al. (2000) | Ferris et al. (2003) | Tobler et al. (2011) | | | | | |
|---|---|---|---|---|---|---|---|---|---|---|---|---|
| | *B. sphaericus 21787* | B. sphaericus 21776 | S. pasteurii ATCC 11859 | S. pasteurii ATCC 6453 | S. pasteurii ATCC 11859 | S. pasteurii ATCC 11859 | S. pasteurii ATCC 11859 | | | | | |
| Temperature (°C) | 30 | 30 | 30 | 25 | 20 | 20 | 20 | 20 | 20 | 20 | 20 | 20 |
| Initial pH | 6.7 | 6.7 | 6.7 | 8.0 | 6.5 | 6.5 | 6.5 | 6.5 | 6.5 | 6.5 | 6.5 | 6.5 |
| $[Ca^{2+}]$ (mM) | 25.2 | 25.2 | 25.2 | 25.2 | 25 | 1.75 | 50 | 200 | 500 | 50 | 200 | 500 |
| [Urea] (mM) | 333 | 333 | 333 | 333 | 333 | 6 | 250 | 250 | 500 | 250 | 250 | 500 |
| [cells] $(OD_{600})$** | 0.015 | 0.014 | 0.014 | 0.010 | 0.072 | 0.070 | 0.03 | 0.03 | 0.03 | 0.07 | 0.07 | 0.07 |
| $k_{urea}$ $(h^{-1})$ | 0.023 | 0.107 | 0.074 | 0.028 | 0.008 | 0.038 | 0.007 | 0.005 | 0.005 | 0.074 | 0.095 | 0.044 |
| $k_{urea}$ $(OD_{600}^{-1}\ h^{-1})$ | 1.526 | 8.020 | 5.251 | 2.800 | 0.111 | 0.543 | 0.250 | 0.180 | 0.180 | 1.065 | 1.363 | 0.630 |
| $k_{urea}$ $(mL\ CFU^{-1}\ h^{-1})$ | 8.52E-09 | 5.30E-08 | 3.22E-08 | 3.00E-08 | 3.73E-10 | 1.81E-9 | 9.86E-10 | 7.13E-10 | 7.13E-10 | 3.56E-09 | 4.56E-09 | 2.11E-09 |
| $k_{precip}$ $(h^{-1})$ | 0.219 | 0.604 | 0.253 | 0.116 | 0.112 | 0.014 | 0.007 | 0.011 | -- | -- | 0.065 | -- |

*Rate coefficients obtained for this strain are not conclusive and should be taken only as a guide for comparison purposes in this study.
**$OD_{600}$ values were converted to a 1 cm path length equivalent where necessary.
--No reported values

