# Peer review of "KINETICS OF CALCITE PRECIPITATION BY UREOLYTIC BACTERIA UNDER AEROBIC AND ANAEROBIC CONDITIONS Andrew C. Mitchell1,2, Erika J. Espinosa-Ortiz2, Stacy L. Parks2,3, Adrienne Phillips2,4, Alfred B. Cunningham2,4, Robin Gerlach2,3 1Department of Geography and Earth Sciences, Interdisciplinary Centre for Environmental Microbiology, Aberystwyth University, SY23 3DB, UK. 2Center for Biofilm Engineering, "

_Biogeosciences, 2018_

## Referee Comment (RC1) · Anonymous Referee #1 · 11 Jan 2019

BG-2018-477

Kinetic of Calcite Precipitation by Ureolytic Bacteria under Aerobic and Anaerobic Conditions

Andrew C. Mitchell, Erika J. Espinosa-Ortiz, Stacy L. Parks, Adrienne Phillips, Alfred B. Cunningham, and Robin Gerlach

The manuscript describes laboratory determinations of the urea hydrolysis (both in the presents and absence of calcite precipitation) and calcite precipitation rates for 3 microorganisms (*S. pasteurii*, *B. sphaericus* 21776, and *B. sphaericus* 2178) at a temperature of 30 °C.  The motivation for the study was the quantification of microbial induced calcite precipitation technology.  The reported experiments were conducted under both aerobic and anaerobic conditions (screening for terminal electron acceptors). The manuscript concluded that rates (as measured by first order rate constants) for both urea hydrolysis and calcite precipitation under aerobic condition were slightly higher for *B. sphaericus* 21776 than for *S. pasteurii*.  Given the scatter among the replicate measure of rate constants (Table SI2.1 CMM+) a more defensible statement would be that there is no apparent difference in rate constants between the two microorganisms.  This statement is further supported by the observation that for the calcium absent experiments the mean urea hydrolysis rate constant (Table SI2.1 CMM-) for *S. pasteurii* is greater than for *B. sphaericus* 21776; the opposite of what was observed for the calcium present experiments.

The manuscript also reported that calcite precipitation inhibits the ability of the microorganisms to hydrolysis urea (presented as the ratios of the hydrolysis rate constants constants) and proposed (based on TEM observations and solid state diffusion calculations) that precipitating calcite entombs some fraction of the microorganisms and that the entombed organisms, while still alive, do not contribute to further hydrolysis.  The manuscript also describes experiments assessing the potential for urea hydrolysis by *S. pasteurii* under anaerobic conditions.  The manuscript concludes that although population growth (as measure by $OD_{600}$) did not occur in the absence of oxygen, urea hydrolysis does occur at rate similar to those for aerobic conditions.  Additionally, growth resume following anaerobic exposure by the re-introduction of oxygen.

The manuscript could be strengthen by additional analysis of the results.   Equations 9 (and 10) represents a first order rate approximation of enzymatically catalyzed urea hydrolysis.  This approach is justified because a) rates were not measured directly (rather urea [or $NH_4$] concertation as a function of time was measured) and b) the integrated rate expression fits the observations (not surprising given the scatter in the observation both within and between experiments).  However enzymatic (urease) kintetics are better describe by the Michaelis-Menten expressions that reduces to $1^{st}$ order kinetics at low urea concentrations and $0^{th}$ order at high urea concentrations.  This results in the apparent rate varying between $1^{st}$ and $0^{th}$ order as a function of urea concentration potentially over the course of a single experiment. With this in mind the normalization process needs to be consider carefully especially in the comparison to results of other studies using different concentration and temperatures.  An alternative approach would be to consider a simple linear regression model of apparent $k_{urea}$ values in term of optical density ($OD_{600}$) and temperature (Kelvin).  Using the results for *S. pasteurii* reported in Table 3

supplemented with the 10 and 15 °C results of Ferris et al. (2004; this is incorrectly reference in both Table 3 and the reference list in the SI as 2003) yields:

log $k_{urea}$ = 34.3 (9.5) + 1.22 (0.47)*log $OD_{600}$ – 10,070 (2,660)/$T_{Kelvin}$

where the values in parentheses represent the standard errors for the parameters.  The fit has an R value of 0.77 (n=13) with the fit parameters having p-values of 0.027 ($OD_{600}$) or less.  If Equation 9 in the manuscript is the proper representation of the kinetics and $OD_{600}$ is the measure of cell concentration (proportional to enzyme concentration), the expect coefficient for log $OD_{600}$ would be 1.   The regression value of 1.22 is less than half the standard error different from 1 suggesting that the assumption of equation 9 are adequate given (or because of?) the scatter in the experimental results, although one cannot exclude the possibility that $k_{urea}$ may also show a dependences on urea concentration (e.g., Michaelis-Menten kinetics).  In addition, the coefficient for the temperature can be used to derive an apparent activation energy thus providing a temperature depended (10 – 30 °C) expression for estimating urea hydrolysis rates.

 A couple of text issues:

Page 7 line 5 –  assuming the reaction is zero order with respect to biomass is not the same as stating that X=0 (if X=0 then d[Urea]/dt in equation 9 is also 0).  Perhaps this section could be revised to consider this a pseudo-first order reaction where the apparent rate constant $k^*_{urea}$ = $k_{urea}$ [X] and d[X]/dt = 0.

Page 8 line 16 (and SI page 4 line 6) – the statement that approximately 25.5 mM of urea need to hydrolyze in order to achieve super-saturation is incorrect.  Figure SI1.1 shows that a DIC concertation of 25.45 mM is required to reach saturation with respect to calcite.  Given that the medium starts with DIC concentration of 25 mM (as NaHCO3; Table SI1.1) only 0.45 (~0.5) mM of urea hydrolysis is required to achieve calcite saturation.

---

## Referee Comment (RC2) · Anonymous Referee #2 · 13 Jan 2019

This is a useful study exploring the kinetics of urea hydrolysis by a group of bacteria and its link to microbial-induced calcite precipitation. Three strains of bacteria are tested (*S. pasteurii*, *B. sphaericus 21776* and *B. sphaericus 2178*), grown at 30C while also varying redox conditions to compare rates of urea hydrolysis between aerobic and anaerobic conditions. Further comparison involves using different terminal electron acceptors in the case of anaerobic experiments, presumably because metabolic pathways based on these TEAs are also known to drive up pH.

The main finding is that only two (*S. pasteurii* and *B. sphaericus 21776*) of the three bacterial strains tested hydrolyse urea and lead to calcite precipitation, and that rates were higher for *B. sphaericus 21776* than *S. pasteurii*, although the differences are not tested statistically. In addition, urea hydrolysis rates are higher when calcium is absent, and biomass (and protein) data is given to show that calcite precipitation arrests the growth of the bacteria through cell entombment, which reduces diffusion of urea to cells. Paradoxically, cell growth is also compromised under anaerobic conditions but this does not appear to affect urea hydrolysis rates.

The study is of wide interest to readers of Biogeosciences and deserves to be published following some minor modifications.

1. There is apparent confusion about the meaning of rate being zero order with respect to biomass being interpreted as X=0, which invalidates both equations 9 and 10. I believe what they mean is that X is constant so the non-normalised rate = $k_{obs}$[Urea], and $k_{urea}$ = $k_{obs}$/[X]. In this context, the calculation of $k_{obs}$ is simply a step towards the calculation of the real $k_{urea}$ and so is not in itself a separate method. Consider this in the same way that observed mineral dissolution rates from solution data are normalised to surface area. The consequence of this is that the comparison with other studies should only be on the biomass-normalised rate constants.

2. In the context of Equation 10, a simpler analysis would have been simply to linearise the function by plotting ln [Urea] against time. I am concerned that other orders were not tested but perhaps it would be useful to provide fitting constraints information (residual sum of squares, r-squared values) for the non-linear fitting presented. Significantly, urea hydrolysis is an enzymatic reaction but this analysis is purely abiotic, while the real rate is likely cell-surface controlled, requiring enzymatic analysis approaches (e.g. Michaelis-Menten).

3. Figure 1 is slightly misleading as it implies urea concentration was measured when it was derived from $NH4^+$ measurements. I wonder if $NH4^+$ data should also be given. Importantly, there is need for an explanation of the drop in Ca concentration in Figure 1C given the limited urea hydrolysis for this bacteria.

4. In line 15, page 8, I am always concerned when control data is "not shown" or presented. It is the only data that gives us confidence that the experimental observations relate to our manipulations rather than chance.

5. In Figure 5, neither of the radial sections show the bright spots which I assume represent the calcium in the cells (arrowed). It would be helpful to clarify why their appearance is orientation-dependent.

---

## Author Comment (AC1) · 19 Mar 2019

We thank the reviewers for their comments on our manuscript. We were very pleased that both reviews were positive suggesting publication with revisions. We have therefore revised the manuscript according to their suggestions. The reviewer comments and responses are shown below:

Reviewer #1 This is a useful study exploring the kinetics of urea hydrolysis by a group of bacteria and its link to microbial-induced calcite precipitation. Three strains of bacteria are tested (S. pasteurii, B. sphaericus 21776 and B. sphaericus 2178), grown at 30C while also varying redox conditions to compare rates of urea hydrolysis be-

[Figure]

tween aerobic and anaerobic conditions. Further comparison involves using different terminal electron acceptors in the case of anaerobic experiments, presumably because metabolic pathways based on these TEAs are also known to drive up pH.

The main finding is that only two (S. pasteurii and B. sphaericus 21776) of the three bacterial strains tested hydrolyse urea and lead to calcite precipitation, and that rates were higher for B. sphaericus 21776 than S. pasteurii, although the differences are not tested statistically. In addition, urea hydrolysis rates are higher when calcium is absent, and biomass (and protein) data is given to show that calcite precipitation arrests the growth of the bacteria through cell entombment, which reduces diffusion of urea to cells. Paradoxically, cell growth is also compromised under anaerobic conditions but this does not appear to affect urea hydrolysis rates.

The study is of wide interest to readers of Biogeosciences and deserves to be published following some minor modifications.

Comment 1: There is apparent confusion about the meaning of rate being zero order with respect to biomass being interpreted as X=0, which invalidates both equations 9 and 10. I believe what they mean is that X is constant so the non-normalised rate = kobs[Urea], and kurea = kobs/[X]. In this context, the calculation of kobs is simply a step towards the calculation of the real kurea and so is not in itself a separate method. Consider this in the same way that observed mineral dissolution rates from solution data are normalised to surface area. The consequence of this is that the comparison with other studies should only be on the biomass-normalised rate constants.

Response: We apologize for the oversight and thank both reviewers for pointing out this issue. This should have stated X = X0 and not X = 0. We modified the text as follows (Page 7, line 11-20):

"Biomass-normalized ureolysis rates were calculated. Firstly, it was assumed that biomass (X) is constant in Eqs. 10 and 11, as performed in other studies of ureolysis kinetics (Cuthbert et al., 2012;Ferris et al., 2004;Mitchell and Ferris, 2005, 2006;Schultz

et al., 2011;Tobler et al., 2011). Secondly, the obtained first order rate coefficients with respect to urea concentration (kurea) were normalized to the biomass concentration by dividing the ureolysis rate coefficient by the biomass at the onset of precipitation (i.e. urea hydrolysis rates were normalized to the absorbance reading of initial biomass, OD600, and CFU mL-1; SI section 2.2), which was equivalent to the initial biomass in each system ($X = X0$). This is an appropriate choice of model, since the biomass analysis indicated that the cell density was constant for the duration of CaCO3 precipitation and was equivalent to the initial biomass in the systems, as presented in the results section. The biomass-normalized ureolysis rates were compared to other parameters previously published (Ferris et al., 2004;Fujita et al., 2000;Stocks-Fischer et al., 1999;Tobler et al., 2011). "

Comment 2: In the context of Equation 10, a simpler analysis would have been simply to linearize the function by plotting ln [Urea] against time. I am concerned that other orders were not tested but perhaps it would be useful to provide fitting constraints information (residual sum of squares, r-squared values) for the non-linear fitting presented. Significantly, urea hydrolysis is an enzymatic reaction, but this analysis is purely abiotic, while the real rate is likely cell-surface controlled, requiring enzymatic analysis approaches (e.g. Michaelis-Menten).

Response: Our previously published work demonstrates that a first-order ureolysis rate model is appropriate in this study, rather than a Michaelis-Menten model, specifically because maximum urea concentrations are 330mM (also see further comments in relation to reviewer 2). We have therefore included a paragraph to justify the use of the first-order ureolysis rate model over the Michaelis-Menten model in our study (Page 7, lines 2-5):

"Indeed, while the Michaelis-Menten model has been used when evaluating ureolysis, studies of ureolysis-induced calcium carbonate precipitation have demonstrated that the first-order ureolysis rate model fits well for urea concentrations of approximately 330 mM or below (Lauchnor et al., 2015; Connolly et al., 2015), which is the concentration

range used in this study."

Comment 3: Figure 1 is slightly misleading as it implies urea concentration was measured when it was derived from NH4+ measurements. I wonder if NH4+ data should also be given. Importantly, there is need for an explanation of the drop in Ca concentration in Figure 1C given the limited urea hydrolysis for this bacteria.

Response: We included a sentence stating that urea concentrations were calculated based on ammonium measurements on page 6, line 14. Because we clarify this, we don't believe it's necessary to present raw NH4+ data which would just be extraneous.

The decrease in Ca concentrations in the B. sphaericus 21787 treatments has now been discussed in much more detail at a number of points in the manuscript.

Specifically on page 8, line 27, we have added: "B. sphaericus 21776 and S. pasteurii exhibit urease activities approximately twice that of B. sphaericus 21787 when Ca is present (Hammes et al., 2003a) supporting our observations of limited ureolysis and delayed CaCO3 precipitation by B. sphaericus 21787. Nevertheless, the decrease in Ca concentrations in the absence of significant urea hydrolysis for B. sphaericus 21787 suggests there was a sufficient carbonate ion concentration in the artificial medium and from ureolysis to initiate some early CaCO3 precipitation. Hammes et al. (2003a) observed B. sphaericus 21787 was also able to precipitate CaCO3 despite lower urease activity in the presence of Ca, suggesting this strain may enhance precipitation via other mechanisms such as enhanced nucleation on cell surfaces or via organic exudates (Mitchell et al, 2006b)."

Then in the urea kinetics section, page 10, line 30, we have added: "B. sphaericus 21787 has been shown to have urease activity about half that of S. pasteurii and B. sphaericus 21776 in the presence of Ca (Hammes et al. 2003a) consistent with the observed low kurea values. However, in the absence of Ca, urease activity for B. sphaericus 21787 is about twice that of B. sphaericus 21776 (Hammes et al., 2003a) which does not support our experimental results. This suggests that while B. sphaericus 21787 has high potential to generate comparable rates of urea hydrolysis to the other strains, under the experimental conditions used in this study, B. sphaericus 21787 exhibits limited ureolytic capabilities."

And also, in relation to the precipitation kinetics on page 11, line 36: "The kprecip for B. sphaericus 21787 was lower than the other strains (0.21 h-1), although as noted, rate constants for this strain are not statistically significant. The lag time for CaCO3 precipitation was 3.3 h for B. sphaericus 21776 and S. pasteurii, which reflects the similar kurea values, and thus the similar time it took to reach CaCO3 saturation and induce precipitation, whereas the longer lag time for B. sphaericus 21776 reflects the significantly lower kurea value."

And in the conclusion, on page 14: "Although B. sphaericus 21787 showed poor ureolysis, some CaCO3 precipitation was observed, suggesting this strain may enhance precipitation via other mechanisms such as enhanced nucleation on cell surfaces or via organic exudates."

Comment 4: In line 15, page 8, I am always concerned when control data is "not shown" or presented. It is the only data that gives us confidence that the experimental observations relate to our manipulations rather than chance.

Response: As per the reviewer's suggestion, we have now included the values of urea concentrations for the control (abiotic experiments). New Figure 1 and caption file attached.

Comment 5: In Figure 5, neither of the radial sections show the bright spots which I assume represent the calcium in the cells (arrowed). It would be helpful to clarify why their appearance is orientation-dependent.

Response: We assume the reviewer refers to Figure 3 since that is the only figure with images in this manuscript. The bright spots are not calcium in the cells. The arrows highlight the breadth of the Ca inclusive precipitate layer which encapsulates

the cell, from the cell wall outwards. We have adjusted the caption in order to clarify this, specifically labelling cell walls (CW) and calcium containing precipitates (CCP) in the images. New Figure 3 and caption file attached.
* * *

---

## Author Comment (AC2) · 19 Mar 2019

We thank the reviewers for their comments on our manuscript. We were very pleased that both reviews were positive suggesting publication with revisions. We have therefore revised the manuscript according to their suggestions. The reviewer comments and responses are shown below:

Reviewer 2: The manuscript describes laboratory determinations of the urea hydrolysis (both in the presents and absence of calcite precipitation) and calcite precipitation rates for 3 microorganisms (S. pasteurii, B. sphaericus 21776, and B. sphaericus 2178) at a temperature of 30 °C. The motivation for the study was the quantification of microbial

[Figure]

induced calcite precipitation technology. The reported experiments were conducted under both aerobic and anaerobic conditions (screening for terminal electron acceptors). The manuscript concluded that rates (as measured by first order rate constants) for both urea hydrolysis and calcite precipitation under aerobic condition were slightly higher for B. sphaericus 21776 than for S. pasteurii.

Given the scatter among the replicate measure of rate constants (Table SI2.1 CMM+) a more defensible statement would be that there is no apparent difference in rate constants between the two microorganisms. This statement is further supported by the observation that for the calcium absent experiments the mean urea hydrolysis rate constant (Table SI2.1 CMM-) for S. pasteurii is greater than for B. sphaericus 21776; the opposite of what was observed for the calcium present experiments.

Response: We agree with the reviewer that the cell number- (i.e. OD600- and CFU/mL-) normalized rate coefficients are not significantly different. We included p-values from two sample t-test comparisons in the manuscript and modified the abstract as follows:

"All bacterial strains showed ureolytic activity inducing CaCO3 precipitation aerobically. First order rate coefficients estimated from the experiments (regardless of whether normalized to biomass concentration or not) demonstrated slightly higher rate coefficients for both ureolysis (kurea) and CaCO3 precipitation (kprecip) for B. sphaericus 21776 compared to S. pasteurii though these differences were not statistically significant. "

And also in the kinetics main section 3.1.3, page 10, line 19: 'S. pasteurii and B. sphaericus 21776 exhibited statistically insignificant differences in kurea values (t-test p-value = 0.27)'.

And page 11, line 34: 'On average, B. sphaericus 21776 had the highest kprecip (0.60 ± 0.34 h-1), although considering its high standard deviation, kprecip for S. pasteurii is not significantly different (kprecip = 0.25 ± 0.02 h-1; t-test p-value 0.21).'

Comment: The manuscript also reports (p 10; LL12-15) that calcite precipitation inhibits the ability of the microorganisms to hydrolyze urea (presented as the ratios of the hydrolysis rate constants constants) and proposed (based on TEM observations and solid state diffusion calculations) that precipitating calcite entombs some fraction of the microorganisms and that the entombed organisms, while still alive, do not significantly contribute to further hydrolysis. The manuscript also describes experiments assessing the potential for urea hydrolysis by S. pasteurii under anaerobic conditions. The manuscript concludes that although population growth (as measured by OD600) did not occur in the absence of oxygen, urea hydrolysis indeed occurred at rate similar to those observed under aerobic conditions. Additionally, growth resumed upon re-introduction of oxygen following anaerobic incubation indicating survival of S. pasteurii for days in the absence of oxygen.

Comment 1: The manuscript could be strengthened by additional analysis of the results. Equations 9 (and 10) represents a first order rate approximation of enzymatically catalyzed urea hydrolysis. This approach is justified because a) rates were not measured directly (rather urea [or NH4] concertation as a function of time was measured) and b) the integrated rate expression fits the observations (not surprising given the scatter in the observation both within and between experiments). However enzymatic (urease) kinetics are better describe by the Michaelis-Menten expressions that reduces to 1st order kinetics at low urea concentrations and 0th order at high urea concentrations. This results in the apparent rate varying between 1st and 0th order as a function of urea concentration potentially over the course of a single experiment. With this in mind the normalization process needs to be consider carefully especially in the comparison to results of other studies using different concentration and temperatures.

An alternative approach would be to consider a simple linear regression model of apparent kurea values in term of optical density (OD600) and temperature (Kelvin).

Using the results for S. pasteurii reported in Table 3 supplemented with the 10 and 15 °C results of Ferris et al. (2004; this is incorrectly reference in both Table 3 and the reference list in the SI as 2003) yields: log kurea = 34.3 (9.5) + 1.22 (0.47)*log OD600 –

10,070 (2,660)/TKelvin where the values in parentheses represent the standard errors for the parameters. The fit has an R value of 0.77 (n=13) with the fit parameters having p-values of 0.027 (OD600) or less. If Equation 9 in the manuscript is the proper representation of the kinetics and OD600 is the measure of cell concentration (proportional to enzyme concentration), the expect coefficient for log OD600 would be 1. The regression value of 1.22 is less than half the standard error different from 1 suggesting that the assumption of equation 9 are adequate given (or because of?) the scatter in the experimental results, although one cannot exclude the possibility that kurea may also show a dependences on urea concentration (e.g., Michaelis-Menten kinetics). In addition, the coefficient for the temperature can be used to derive an apparent activation energy thus providing a temperature depended (10 – 30 °C) expression for estimating urea hydrolysis rates.

Response: We agree with the reviewer that Michaelis Menten-type of kinetics are most appropriate for describing enzyme kinetics over a wide range of concentrations and that the observed catalysis rate (v) indeed would be concentration-dependent. However, it has been demonstrated in the literature repeatedly that a first order first-order ureolysis rate model is appropriate in this study, rather than a the Michaelis-Menten model, specifically because urea concentrations are at concentration of 330mM or below. We have therefore included a paragraph justifying the use of the first-order ureolysis rate model over the Michaelis-Menten model in our study (Page 7, lines 2-5):

"Indeed, while the Michaelis-Menten model has been used when evaluating ureolysis, studies of ureolysis-induced calcium carbonate precipitation have demonstrated that the first-order ureolysis rate model fits well for urea concentrations of approximately 330 mM or below (Lauchnor et al., 2015; Connolly et al., 2015), which is the concentration range used in this study."

A first order approach indeed facilitates communication of the observed results since only one parameter (the first order kinetic rate coefficient) has to be compared and the overall goal of this manuscript was to compare the achievable urea hydrolysis and calcite precipitation rates for the three different strains (S. pasteurii, B. sphaericus 21776, and B. sphaericus 21787) and for S. pasteurii under aerobic and anaerobic conditions.

As reviewer 2 points out here, Michaelis Menten-type kinetics can be approximated with first or zero kinetics for concentration ranges well below or above the half saturation constant (kM), respectively. As stated above, there is ample evidence in the literature (e.g. Lauchnor et al., 2015; Connolly et al., 2015) that first order kinetic approaches describe the kinetic behavior of urea hydrolysis for urea concentrations at 330mM or below quite well (see also Mitchell and Ferris, 2005; 2006a, b; Ferris et al 2004; Tobler et al., 2011; Cuthbert et al 2011) and the overall goal of this manuscript was to compare the achievable urea hydrolysis and calcite precipitation rates for the three different strains (S. pasteurii, B. sphaericus 21776, and B. sphaericus 21787) and for S. pasteurii under aerobic and anaerobic conditions. We indeed agree that additional analyses (temperature dependencies, OD dependencies, etc.) could be wrapped into a more widely valid kinetic equation but we are uncomfortable doing so at the current time since there are too many uncertainties regarding the exact manner in which data by other groups (Tobler et al 2011; Ferris et al., 2004) were acquired.

We also in response to reviewer 1's comment 1 addressed the normalization to biomass concentration and outlined the changes made in this section.

Ferris et al. (2004) reference has been corrected in both, Table 3 and the reference list in the supplementary information.

Comment 2: Page 7 line 5 – assuming the reaction is zero order with respect to biomass is not the same as stating that X=0 (if X=0 then d[Urea]/dt in equation 9 is also 0). Perhaps this section could be revised to consider this a pseudo-first order reaction where the apparent rate constant k*urea = kurea [X] and d[X]/dt = 0.

Response: As stated in the response to comment 1 of reviewer 1: We apologize for the oversight and thank both reviewers for pointing out this issue. This should have stated X = X0 and not X = 0. As outlined in our response to reviewer 1, we have now modified

the text as follows (Page 7, line 11) :

"Biomass-normalized ureolysis rates were calculated. Firstly, it was assumed that biomass (X) is constant in Eqs. 10 and 11, as performed in other studies of ureolysis kinetics (Cuthbert et al., 2012;Ferris et al., 2004;Mitchell and Ferris, 2005, 2006;Schultz et al., 2011;Tobler et al., 2011). Secondly, the obtained first order rate coefficients with respect to urea concentration (kurea) were normalized to the biomass concentration by dividing the ureolysis rate coefficient by the biomass at the onset of precipitation (i.e. urea hydrolysis rates were normalized to the absorbance reading of initial biomass, OD600, and CFU mL-1; SI section 2.2), which was equivalent to the initial biomass in each system (X = X0). This is an appropriate choice of model, since the biomass analysis indicated that the cell density was constant for the duration of CaCO3 precipitation and was equivalent to the initial biomass in the systems, as presented in the results section. The biomass-normalized ureolysis rates were compared to other parameters previously published (Ferris et al., 2004;Fujita et al., 2000;Stocks-Fischer et al., 1999;Tobler et al., 2011)."

Comment 3: Page 8 line 16 (and SI page 4 line 6) – the statement that approximately 25.5 mM of urea need to hydrolyze in order to achieve super-saturation is incorrect. Figure SI1.1 shows that a DIC concertation of 25.45 mM is required to reach saturation with respect to calcite. Given that the medium starts with DIC concentration of 25 mM (as NaHCO3; Table SI1.1) only 0.45 ( 0.5) mM of urea hydrolysis is required to achieve calcite saturation.

Response: We agree with the reviewer and truly appreciate the thorough review. Based on the modeling It indeed only takes an additional 0.45 mM of urea hydrolyzed in order to reach a SI = 0 for calcite since the medium already contains 25 mM bicarbonate. – the overall bicarbonate concentration is predicted to be 25.5 mM at that point. – We corrected the statement in the manuscript and in the supplementary information: "Geochemical modelling suggested that no CaCO3 precipitation should occur in the absence of ureolysis and that approximately 0.45 mM of urea would have had to be
hydrolyzed to achieve supersaturation and for CaCO3 precipitation to commence (see SI1.4)."

We thank the reviewers again for their valuable comments and look forward to hearing back.

---

## Author Response (AR1)

**Authors response to editor's comments**

**Manuscript number:** bg-2018-477

**Title:** Kinetics of calcite precipitation by ureolytic bacteria under aerobic and anaerobic conditions

**Journal:** Biogeochemistry

We thank the editor for her comments on our manuscript. We have revised the manuscript in accordance with reviewer responses (as addressed in our responses via the interactive review 20/03/19), and according to editor suggestions from 31/03/19. These responses to the editor's comments are shown below:

**Comment 1:** Pg. 5 l. 2: S. pasteurii (ATTC 11859) –is this ATCC?

*Response: Yes, it should be ATCC. We have corrected this in the modified manuscript.*

**Comment 2:** Pg. 5, l. 29: use TEA here since it was already abbreviated

*Response: We agree with the editor and used TEA instead of terminal electron acceptor in the the revised version of the manuscript.*

**Comment 3:** Pg. 5, l. 30-32: this reads awkwardly—you didn't make the nitrate solution from 1M NaNO3 but the solution was 1M NaNO3. Suggest revising to "…NO3- was added from a solution of 1M NaNO3; ii) a concentrated SO42- solution was made by combining 1M Na2SO4 and 1M Na2S, Na2S was added to quench any residual oxygen and make SO42- reduction possible; and iii) Fe3+ was added from a stock solution of 50 mM Fe(III) citrate made as previously described."

**Response: T**he sentence now reads: *Concentrated stock solutions of each TEA were made in the anaerobic chamber and filter sterilized: i) a 1M solution of $NaNO_3$; ii) a concentrated $SO_4^{2-}$ solution, made by combining 1M $Na_2SO_4$ and 1M $Na_2S$, where $Na_2S$ was added to quench any residual oxygen and make $SO_4^{2-}$ reduction possible; and iii) a stock solution of Fe(III) citrate, using 50 mM Fe(III) citrate as previously described (Gerlach et al., 2011).*

**Comment 4:** Pg. 6, l. 2: do you mean anaerobic control here?

*Response: No. We were referring to the comparative aerobic experiments including TEAs, as shown in Figure 4 (circles). We have corrected this in the revised version of the manuscript to read;*

"Comparative aerobic control experiments were also performed with CMM- media including 10 mM $NO_3^-$, $SO_4^{2-}$, or $Fe^{3+}$ and inoculated with 1 mL of *S. pasteurii* in 150 mL serum bottles."

**Comment 5:** Pg. 7, l. 16: please correct sub- and superscripts

**Response:** *We corrected the sub- and superscripts in the revised version of the manuscript.*

**Comment 6:** Pg. 13, l. 6: growth and pH seem pretty similar between the nitrate and Fe treatments. Its not clear why nitrate was preferentially chosen

**Response:** *We agree there are not significant differecnes between the different TEAs. We have therefor altered the justification in section 3.2.2. to read;*

"After the screening experiments with different TEAs, studies were performed to determine the kinetics of ureolysis and $CaCO_3$ precipitation in the absence of oxygen. Since there were similarly low levels of growth in the initial anaerobic screening experiments with different TEAs and no TEA (Figure 4), kinetic experiments in the absence of oxygen were conducted with no TEA as well as with $NO_3^-$ as a potential TEA (Figure 5)."

**Comment 7:** Pg. 14, l. 16: correct italics

**Response:** *The italics have been corrected in the revised version of the manuscript.*

**Comment 8:** Figure 4: Give abbreviation TEAs in the figure caption. There are only open data points for some panels, why? Why have OD measurements and not pH after the bottles were opened?

**Response:** *We included the abbreviation TEAs in the figure caption in the revised version of the* manuscript. We have also modified Figure 4 and included the open data points for the pH graphs. The corrected figure has been included in the modified version of the manuscript and for reference below.

[Figure]

**Comment 9:** Figure 5 caption: correct spelling of circle.

*Response: We corrected the figure caption in the revised version of the manuscript.*